# Pomegranate (*Punica granatum* L.) and Metabolic Syndrome Risk Factors and Outcomes: A Systematic Review of Clinical Studies

**DOI:** 10.3390/nu14081665

**Published:** 2022-04-16

**Authors:** Lucas Fornari Laurindo, Sandra Maria Barbalho, Alexis R. Marquess, Annik Ianara de Souza Grecco, Ricardo de Alvares Goulart, Ricardo José Tofano, Anupam Bishayee

**Affiliations:** 1Department of Biochemistry and Pharmacology, School of Medicine, University of Marília, Marília 17525-902, SP, Brazil; lucasffffor@gmail.com (L.F.L.); annikgreco@yahoo.com.br (A.I.d.S.G.); rtofano@uol.com.br (R.J.T.); 2Postgraduate Program in Structural and Functional Interactions in Rehabilitation, University of Marília, Marília 17525-902, SP, Brazil; ricardogoulartmed@hotmail.com; 3Department of Biochemistry, School of Food and Technology of Marilia, Marília 17500-000, SP, Brazil; 4College of Osteopathic Medicine, Lake Erie College of Osteopathic Medicine, Bradenton, FL 34211, USA; amarquess66829@med.lecom.edu

**Keywords:** *Punica granatum*, pomegranate, body weight, hyperglycemia, hypertension, metabolic syndrome

## Abstract

Pomegranate (*Punica granatum* L.) can be considered a multipurpose medicinal and dietary plant due to its anti-inflammatory and antioxidant actions. Pomegranate can be used to prevent or treat metabolic syndrome (MetS) risk factors. Although previously published reviews addressed the effects of pomegranate on different diseases, there is no systematic review that exclusively focuses on clinical trials related to all MetS-related risk factors. In view of this limitation, the objective of this up-to-date, comprehensive, and systematic review is to critically evaluate the potential of pomegranate (*P. granatum*) on various MetS risk factors on the basis of clinical studies. PubMed, EMBASE, MEDLINE, Google Scholar, COCHRANE, and Clinical Trials.gov databases were searched on 15 October 2021. The Preferred Reporting Items for a Systematic Review and Meta-Analysis guidelines were followed, and the bias risk evaluation was performed according to the Cochrane Handbook for Systematic Reviews of Interventions. We identified 5683 studies in the databases. After removing the duplicates, 3418 studies remained. Of these, 147 studies met the eligibility criteria, and finally, only 20 were included in the qualitative analysis. The included studies suggest that pomegranate can be beneficial to reduce body weight, blood pressure, glycemia, triglycerides, total cholesterol, and low-density lipoprotein cholesterol. Moreover, it can augment high-density lipoprotein cholesterol levels and improve insulin resistance. Although relevant effects were observed, additional well-designed clinical trials are needed to determine the correct formulations and doses to be used to prevent or treat MetS components.

## 1. Introduction

Since ancient times, people have appreciated natural agents, and plant-derived substances have been used as a source for food and medicine due to their numerous health properties [1,2]. Pomegranate (*Punica granatum* L.) can be considered a multipurpose medicinal and dietary plant [3,4]. This species is part of the Punicaceae family and corresponds to being one edible fruit that has been known since antiquity. Native to the center of the Asian continent (specifically to Iran), the pomegranate represents a deciduous shrub or small tree that finds extensive use as a medicinal agent [3,4,5]. This plant has been introduced to other continents, such as America and Africa, thus being cultivated in these areas. The anti-inflammatory and antibacterial properties of pomegranate are well known, and these effects are implicated in the prevention and treatment of various human diseases using this superfruit [6,7].

The most known part of the *P. granatum* plant is the fruit, which is most used for baking, cooking, and producing beverages [8,9]. This fruit has been used across generations due to its anti-inflammatory, antioxidant, and antitumor effects, alongside its neuroprotection and cardiovascular protection properties [10,11]. *P. granatum* consumption is also associated with antidiabetic effects and it influences other aspects of the diagnostic components of metabolic syndrome (MetS) [12,13].

The globalized world is beneficial to many aspects of humans’ lives, bringing comfort and facilities to the day-after-day routine. Besides that, the consequent lifestyle changes are considered in many aspects to cause disorders. MetS is a complex metabolic disorder, driving health and socioeconomic challenges worldwide, and is considered an epidemic [14,15]. Affecting both sexes, MetS is associated with many cardiometabolic risk factors, and its prevalence in several countries reaches 30% of the total population. The importance of this metabolic condition remains that this disease is significantly related to an increase in cardiovascular risk, leading to cardiovascular diseases, which are considered the chronic degenerative disorders associated with high morbidity and mortality [15,16,17].

MetS is represented by a cluster of different metabolic abnormalities that include hypertension, abdominal or central obesity, insulin resistance (IR), hyperglycemia, and an atherogenic profile of dyslipidemia (low levels of high-density lipoprotein cholesterol (HDL-c) and high levels of serum triglycerides) [14,15]. This metabolic disorder is a global concern, principally due to its high prevalence among the adult population [16]. Although the pathophysiology of this syndrome involves acquired and genetic factors, other risk factors include a positive family history, low socioeconomic status, physical inactivity, history of diabetes, fatty liver disease, unhealthy alimentary habits, use of specific drugs, the aging process, alcoholism, and smoking [17,18]. IR and obesity are the key points of MetS pathophysiology that primarily include inflammation, neurohumoral activation, and IR [19,20]. In addition to the pro-inflammatory adipokines, stimulation of the renin–angiotensin–aldosterone system and the pro-oxidative state can contribute to chronic inflammation [21,22,23,24]. Figure 1 summarizes the main pathophysiological aspects of MetS and the effects of pomegranate on these dysfunctional risk factors.

*P. granatum* contains carbohydrates, fiber, proteins, minerals, and vitamins, alongside other nutrients, that are important for preventing MetS [13]. The fruit and other components of the plant contain numerous bioactive compounds, such as phenolic acids, hydrolysable tannins, condensed tannins, and flavonoids, as well as other types of bioactive constituents [25,26,27,28] that can exert antimicrobial [29], anticancer [30], antioxidant [26,31], and anti-inflammatory activities [28]. The plant can also affect glucose and lipid metabolisms [32,33,34], as well as other metabolic conditions, such as obesity [27,34] and IR [35]. Effects of pomegranate on cardiovascular health have also been reported, together with antiosteoporosis properties [35,36]. Table 1 shows the bioactive components of pomegranate and their effects related to MetS on the basis of preclinical and clinical studies.

Several research groups previously reviewed the effects of pomegranate in various metabolic diseases. Medjakovic and Jungbauer [60] summarized the effects of pomegranate fruit on MetS on the basis of in vitro and in vivo findings. Hou et al. [32] reviewed the preclinical effects of pomegranate on lipid profiles. Jandari et al. [61] provided an overview of pomegranate effects on diabetic patients. Eghbali et al. [25] performed a simple review of clinical trials that evaluated the effects of pomegranate on oral cavity disorders, stomatitis, diabetes, cardiovascular and neoplastic diseases. The study of Akaberi et al. [62] summarized the therapeutic effects of pomegranate and its active constituents in metabolic disorders on the basis of in vitro and in vivo studies. Although these studies are interesting, they are not systematic reviews. Asgary et al. [7] published a systematic review and meta-analysis focused on pomegranate’s actions on vascular adhesion factors. Although previously published reviews addressed the effects of pomegranate on different diseases, there is no systematic review that exclusively focuses on clinical trials related to all MetS-related risk factors. In view of this limitation, the objective of this up-to-date, comprehensive, and systematic review is to critically evaluate the potential of pomegranate (*P. granatum*) on various MetS risk factors on the basis of clinical studies.

## 2. Literature Search Strategy and Study Selection

The search for this review was performed on 15 October 2021 and was carried out by two authors (L.F.L. and S.M.B.). For this work, only clinical studies published in English were selected using PubMed, EMBASE, MEDLINE, Google Scholar, COCHRANE, and Clinical Trials.gov databases. The descriptors used were *P. granatum* or pomegranate and metabolic syndrome or syndrome X, hypertension or blood pressure, lipid profile or lipids or triglycerides or high-density lipoprotein cholesterol, obesity or body weight or waist circumference, insulin resistance or diabetes or glycemia, and clinical trials or trials. All these descriptors helped the authors to identify published clinical trials that associated the consumption of any part or extract of *P. granatum* or pomegranate with ameliorations of the components of MetS. The guidelines of the 2020 Preferred Reporting Items for a Systematic Review and Meta-Analysis (PRISMA) were followed to perform this systematic review [63]. The inclusion criteria considered randomized controlled trials (RCTs) that included double-blind and single-blind studies, placebo-controlled trials, prospective and crossover studies, triple-blind trials, parallel-group studies, and open-label trials that were published in English. The exclusion criteria comprised of in vitro studies, in vivo studies (such as animal experiments), editorials, review studies, posters presentations, conference abstracts, and case reports. The risk of bias evaluation of the RCTs involved with this systematic review was evaluated according to the Cochrane Handbook for Systematic Reviews of Interventions [64].

## 3. Result Findings

Figure 2 represents a scheme to depict a literature search and study selection process. We identified 5683 studies in the databases that were searched. After removing the duplicates, 2458 studies remained. Of these, 147 studies met the eligibility criteria and, after applying the inclusion and exclusion criteria, only 20 were included in the qualitative analysis. From the 20 studies selected and included in this systematic review, seven were from Iran [10,12,65,66,67,68,69], four were from the United States [70,71,72,73], three were from the United Kingdom [74,75,76], one was from Mexico [77], one was from Spain [78], one was from Greece [79], one was from Serbia [80], one was from Israel [81], and one was from South Korea [82]. In the included studies, the respective authors used different formulations derived from pomegranate to treat the participants: pomegranate juice [10,12,65,68,69,71,72,73,75,76,77,78,79,80,81], pomegranate extract [68,70,73,74], pomegranate juice enriched with probiotic microorganisms [67], diluted pomegranate juice [66,78], and pomegranate vinegar [82]. The studies did not report serious adverse effects and high drop-outs rates. However, two studies had more than 20% drop-outs [78,81].

From the 20 included RCTs, 828 participants were evaluated in this systematic review. From the total participants, 21 were diagnosed with hypertension, 136 were women diagnosed with polycystic ovary syndrome (PCOS), 175 were healthy subjects, 19 had severe carotid artery stenosis and were symptomatic, 146 were obese or overweight, 112 were diagnosed with type 2 diabetes mellitus (T2DM), 45 had ischemic coronary heart disease and myocardial ischemia, 199 had end-stage renal disease (ESRD) or needed hemodialysis treatment, and 23 were diagnosed with MetS.

The included studies have shown that the treatment with pomegranate, pomegranate extracts, or pomegranate-derived substance was able to improve cardiometabolic risk factors, such as fasting blood glucose and glycemic levels, systolic and diastolic blood pressures, lipid profiles, and body weight, which are the main diagnostic aspects of MetS (Table 2). Total cholesterol (TC), low-density lipoprotein cholesterol (LDL-c), HDL-c, serum triglycerides (TG), body mass index (BMI), homeostasis model of assessment insulin resistance (HOMA-IR), systolic blood pressure (SBP), and diastolic blood pressure (DBP), fasting blood glucose levels, and serum insulin levels were assessed from the studies in which these risk factors were disponible. Additionally, in many clinical trials, the inflammatory and oxidative information of the participants were available. These were mainly interleukin-6 (IL-6) levels, C-reactive protein (CRP), high-sensitivity CRP (hs-CRP) levels, malondialdehyde (MDA) levels, and total antioxidant capacity (TAC) levels.

Table 3 shows the description of the bias observed in the included studies according to the Cochrane Handbook for Systematic Reviews of Interventions [64]. Undertaking this methodological research tool, we were able to examine if certain sources of bias could show a substantial impact on the quality of the included clinical trials, diminishing the risk of bad systematic review quality.

## 4. Discussion

Several clinical trials investigated the effects of *P. granatum* in metabolic conditions in humans. These studies are shown in Table 2. The potential bias is described in Table 3. In the following sections, we briefly discuss various RCTs.

Abedini et al. [66] showed that the use of concentrated pomegranate juice can produce beneficial actions on blood pressure, TG, and HDL-c levels. One limitation of this study is the non-double-blinded assessments. Additionally, this study presented a small sample size, which can interfere with the results.

Esmaeilinezhad et al. [67] evaluated the effects of pomegranate juice added to symbiotics on reducing cardiovascular risk factors of women diagnosed with PCOS. The results showed that the pomegranate intervention improved metabolic, oxidative, hemodynamic, and inflammatory characteristics of the treated women. This study was triple-blind, and the sample size was calculated.

The trial of Barati Boldaji et al. [10] assessed the actions of pomegranate juice intake on cardiometabolic risk factors of hemodialysis patients. The study was not double-blind, and there is a lack of a constant control group due to ethical approvements. These ethical consensuses obligated the authors of this study to change the participants between control and intervention groups after a washout period of 4 weeks. This study also presents a small sample size and is not double-blind.

Sohrab et al. [12] conducted a trial to evaluate the actions of pomegranate juice on blood pressure and lipid profiles of individuals affected by T2DM. The main limitation of this study was the short duration of the intervention. The intervention brought benefits on blood pressure rates, although there were no benefits to the individual lipid profiles. Information about possible alterations of fasting blood sugar was not mentioned in the study. Six intervention group participants reported the use of antihypertensive drugs, in turn of seven in the control group. Twenty-seven subjects of the intervention group reported using at least one oral hypoglycemic drug, in turn of 25 in the control group. Seven intervention group participants reported the use of lipid-lowering drugs, in turn of five in the control group. These findings can be potentially biased to the results of the study. Another limitation is that this study was not double-blind, and the duration of the trial was short.

Kojadinovica1 et al. [80] conducted a randomized study to investigate the effects of pomegranate juice on lipid metabolism of women diagnosed with MetS. The sample size was 23, which can be considered a limitation of this study due to its small size. Another limitation lies in the fact that only women were evaluated. Additionally, this clinical trial was not double-blind, and the placebo group received water instead of something that was more similar to pomegranate juice.

Stockton et al. [74] investigated the effects of pomegranate extract capsule intake on blood pressure and anthropometry parameters. The results showed positive effects on DBP and non-significant effects on body-weight parameters and SBP of the participants who received pomegranate intervention.

Manthou et al. [79] investigated the effects of pomegranate juice consumption on biochemical and blood parameters. The sample size was small but was calculated. The results showed positive effects on erythropoiesis and a non-significant impact on lipid profiles and glucose levels. One limitation is that this study is not a double-blind trial. Furthermore, there is a lack of demographic information of the participants of each group (intervention and control).

The trial of Hosseini et al. [68] evaluated the effects of pomegranate extract intake on inflammatory parameters of obese and overweight individuals. One relevant limitation of this trial is the small duration.

Fuster-Muñoz et al. [78] conducted a trial to evaluate the actions of pomegranate juice consumption on circulating cytokines and OS parameters of athletes. This study was conducted only with men and had a high drop-out rate. Although the intervention occurred in a short time (21 days), changes in malondialdehyde and carbonyls were observed. Besides this, the control group did not consume a placebo, but rather a small piece of the proper pomegranate fruit.

Rivara et al. [73] performed a study to evaluate the actions of pomegranate supplementations in patients submitted to hemodialysis (patients diagnosed with end-stage renal disease). The results of this trial showed that the interventions were well tolerated. However, none of them demonstrated significant benefits to lipid profiles or inflammatory and OS parameters. One limitation of this trial is the small size of the studied sample, which can limit the study results. Another limitation is that this study is not double-blind.

Wu et al. [70] discussed the effects of supplementation of a pomegranate extract on physical function and cardiovascular risk factors of hemodialysis patients. It was found that the treatment with pomegranate extract reduced SBP and DBP and increased antioxidant capacities. In the pomegranate extract supplementation group, paraoxonase-1 activity was increased by more than 26. Information about fasting blood glucose rates was not found in this study. At baseline, the characteristics of blood pressures between intervention and control groups were very different. Another limitation of this trial is that it used a small sample, which can limit the study conclusions. Moreover, the study was not double-blind.

Shema-Didi et al. [81] conducted a trial with hemodialysis participants. After the intervention time, HDL-c, TG SBP, DBP, and pulse pressure rates were improved. One strong point of this trial is the long-term interventional period, although the intervention with polyphenolic-rich pomegranate juice was performed only three times weekly. The high dropout rate observed in this study and the resulted small sample size are all limitations that can limit the study results.

The trial performed by Asgary et al. [65] evaluated the actions of pomegranate juice in blood pressure, endothelial function, inflammation, and lipid profiles of hypertensive subjects. The results showed hypotensive effects, in addition to effects ameliorating endothelial function. The small sample size, the percentage of women participants in both groups (more than 70% in each group), and the non-double-blinded assessments are all characteristics that may interfere with the results of the study. Moreover, the intervention period was small (2 weeks).

Sohrab et al. [69] evaluated the effects of pomegranate juice intake on participants diagnosed with T2DM. The sample size of this RCT was calculated, but there were six drop-outs. This study demonstrated that pomegranate juice can exert anti-inflammatory effects on participants diagnosed with T2DM due to measurements of inflammatory biomarkers. This intervention presented a complete table of dietary factors involved at baseline and in the participants’ final intervention. OS parameters could have been measured, and this is a limitation of the study.

The trial performed by Park et al. [82] evaluated the actions of pomegranate vinegar on visceral fat accumulation. The intervention led to a reduction in the visceral accumulation of fat in the treated participants and enhanced AMPK (AMP-activated protein kinase) phosphorylation, which only improved adiposity levels. One limitation of this trial is that only women were included, which can limit its conclusions.

Tsang et al. [76] performed a trial to evaluate the actions of pomegranate juice on the urinary glucocorticoid homeostasis model assessment of insulin resistance (HOMA-IR) and blood pressure of volunteers. This trial was not double-blind, and the sample was small. Furthermore, some data are missing, such as for weight and BMI of the control group.

Lynn et al. [75] evaluated the effects of pomegranate juice on pulse wave velocity and blood pressure parameters. There was a sample size calculation, but the design was open label. The adverse effects were not reported. Possible modifications of lipid profiles and glucose parameters from baseline to the end of the intervention were not demonstrated.

González-Ortiz et al. [77] conducted a trial to evaluate the effects of pomegranate juice consumption on insulin parameters of obese participants. This study had a calculated sample size and was double blind. There was a lack of demographic information about the participants between the groups. No adverse effects were reported.

Sumner et al. [71] investigated the effects of pomegranate juice consumption on myocardial perfusion of individuals diagnosed with ischemic coronary heart disease and myocardial ischemia. Participants of both groups presented use of lipid-lowering, antihypertensive, and antidiabetic drugs, in addition to the use of nitrates by some participants. After the interventional period, the stress-induced extent of ischemia was diminished in the treated group and increased in the control. In addition, hypertensive and hyperlipidemic subjects were more prevalent in the pomegranate juice group than the control. Moreover, hypertensive and hyperlipidemic subjects were more prevalent in the pomegranate juice group than the control.

Aviram et al. [72] conducted a trial to evaluate the effects of pomegranate juice on participants diagnosed with carotid artery stenosis. This study was initially designed for one year, but some participants agreed to continue the intervention along for up to 3 years. This 3-year consumption reduced standard carotid intima-media thickness, LDL oxidation, and blood pressure. The improvements in blood pressure were accentuated in the first 1 year of intervention. This study was not double-blind, and there is a lack of confident prognostic and demographic information about the participants, which can be limitations of the study. Moreover, adverse effects were not well reported.

Various bioactive actions of *P. granatum* can be useful in the prevention or the reduction of several health disorders, including reduction of MetS risk factors, since they can reduce inflammation and OS, thus preventing damage to cell components, reducing insulin resistance and improving dyslipidemia and fat accumulation [38,39,49,58,59,83].

Moreover, *P. granatum* possesses flavonoids, tannins, ascorbic acid, α-tocopherol, and β-carotene that also can be related to the improvement of the oxidative damage observed in patients with the above-mentioned MetS risk factors. In the peel and in the juice, it is possible to identify the presence of punicalagin, a potent antioxidant compound. The peel also contains catechins, gallic acid, ellagic acid, punicalagin, ellagitannins, and punicalin. These bioactive compounds can decrease the risk of development of endothelial dysfunction and cardiovascular outcomes [38,39,49,58,59,83].

*P. granatum* and its BCs are well tolerated and exert health effects both in human and animal organisms. The plant and its derivatives and extracts ameliorate some altered metabolic affections, such as diabetes, hypertension, and dyslipidemia. Added to that, pomegranate can be used to protect against hypertension. However, the results of RCTs that assess these effects are diverse. Nevertheless, the conduction of other RCTs is extremely necessary to the correct establishment of effective doses, treatment periods, and pharmaceutical presentations needed for the use of pomegranate to treat and prevent MetS diagnostic components.

Figure 3 summarizes the most relevant findings of the included RCTs. Pomegranate constituents have been responsible for reducing systolic and diastolic blood pressure, as well as body weight. Additionally, the use of pomegranate has also been linked to reduced dyslipidemia and improved HDL-c levels. Lastly, pomegranate treatments improved glycemia.

## 5. Conclusions

On the basis of the results as presented in this systematic review, we can conclude that the use of pomegranate can be beneficial in exerting reductions of MetS risk factors, such as body weight, blood pressure, glycemia, triglycerides, LDL-c, and total cholesterol. Moreover, it can augment HDL-c levels. Although these effects have been observed, more clinical studies are necessary to determine the correct formulations and doses that should be used to prevent or treat MetS components.

## Figures and Tables

**Figure 1 nutrients-14-01665-f001:**
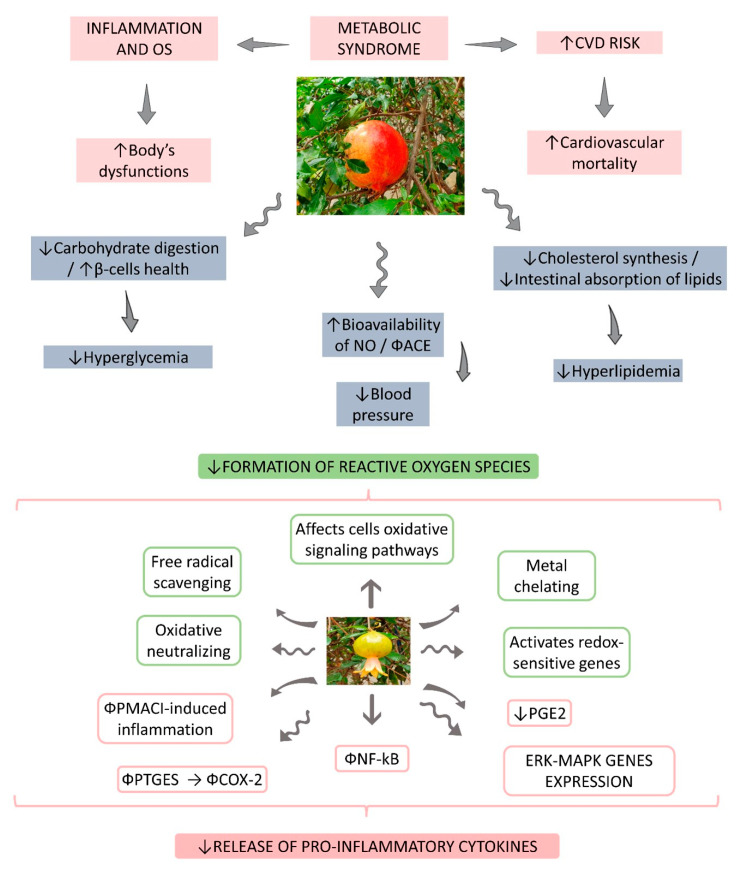
Metabolic syndrome and pomegranate: a blocking relationship. ↑, increase; ↓, decrease; Φ, impairment; β, beta; ACE, angiotensin converting enzyme; COX-2, cyclooxygenase; ERK, extracellular signal-regulated kinase; MAPK, mitogen-activated protein kinase; NF-κB, nuclear factor-κB; NO, nitric oxide; PMACI, phorbol 12-myristate 13-acetate and calcium ionophore; PGE2, prostaglandin E2; PTGES, prostaglandin E synthase.

**Figure 2 nutrients-14-01665-f002:**
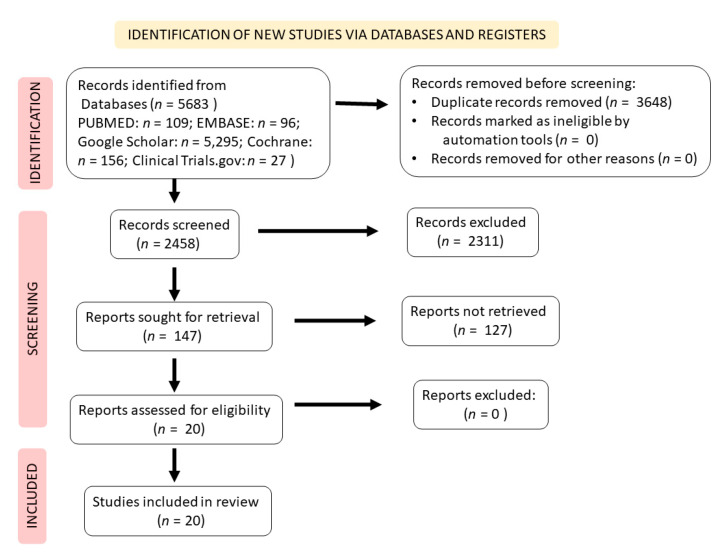
Flow diagram showing the literature search and study selection criteria. Analysis of the literature and writing of the manuscript were performed in accordance with the Preferred Reporting Items for Systematic Reviews and Meta-Analyses (PRISMA) guidelines [63].

**Figure 3 nutrients-14-01665-f003:**
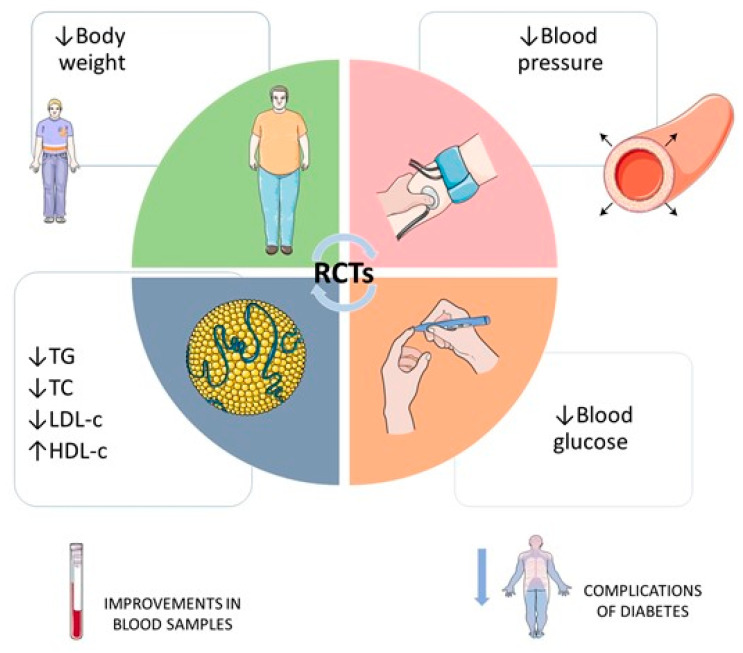
Pertinent findings of the included randomized clinical trials. ↑, increase; ↓, decrease; HDL-c, high-density lipoprotein cholesterol; LDL-c, low-density lipoprotein cholesterol; RCTs, randomized clinical trials; TC, total cholesterol; TG, triglycerides.

**Table 1 nutrients-14-01665-t001:** Bioactive compounds of *P. granatum* and its actions on the risk factors related to MetS.

Bioactive Compounds	Plant Parts	Molecular Structures	Effects	References
Gallic acid	Peel, juice, flower, seeds, and fruit	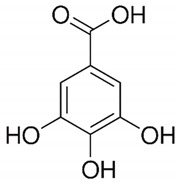	Antidiabetic, anti-inflammatory, and antioxidant	[36,37,38,39,40,41,42,43,44]
Ellagic acid	Peel, juice, fruit, flower, and seeds	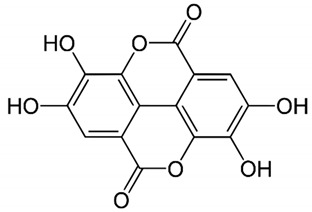	Antidiabetic, anti-obesity, anti-inflammatory, and antioxidant	[32,34,36,37,38,39,40,45,46,47,48]
Quercetin	Peel and seeds	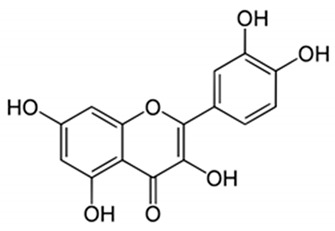	Antidiabetic and anti-inflammatory	[8,36,37,38,39,40]
Punicalin	Peel, juice, and fruit	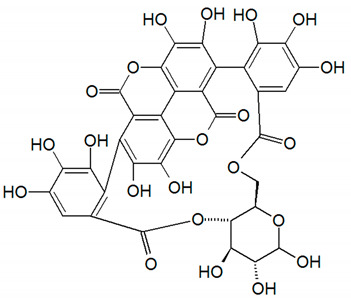	Antidiabetic and antioxidant	[36,37,38,39,40,48,49]
Epicatechin	Peel	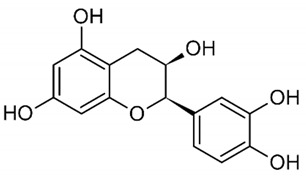	Anti-inflammatory	[36,37,39,48]
Tannic acid	Peel	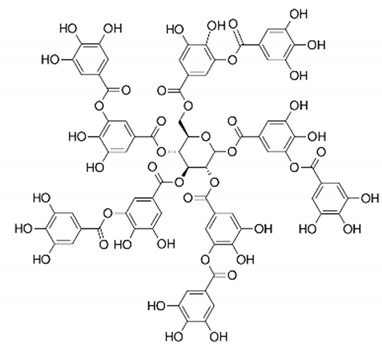	Anti-obesity and antioxidant	[34,50,51,52]
Punicalagin	Peel, flower, seeds, juice, and fruit	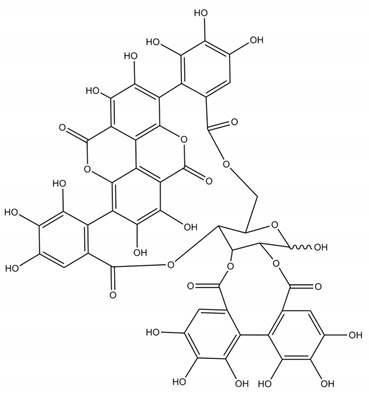	Antidiabetic, antioxidant, and anti-inflammatory	[8,32,36,37,38,39,46,49]
Urolithin A	Polyphenol ellagitannin–gut microbial-derived metabolite	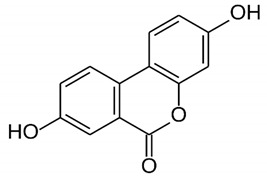	Anti-obesity and anti-inflammatory	[53,54,55,56,57]
Urolithin B	Polyphenol ellagitannin–gut microbial-derived metabolite	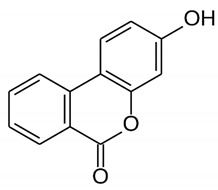	Anti-obesity and antioxidant	[53,55,56]
Anthocyanins	Fruit	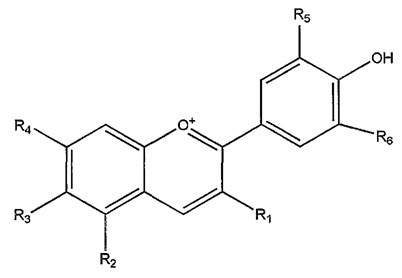	Antioxidant	[38,49,58,59]

**Table 2 nutrients-14-01665-t002:** The effects of *P. granatum* on various components of MetS.

Type of the Study	Country of the Study	Interventions	Lipid Profile	Body Weightor Obesity	Diabetes or IR	Blood Pressure	Adverse Effects	Reference
Randomized clinical trial with 42 women participants (18–40 years) diagnosed with PCOS.	Iran	Participants were randomized into two groups: intervention (*n* = 21, 45 mL/day of concentrated pomegranate juice added to 180 mL of water for 8 weeks) and control (*n* = 21).	Reduction of TG (*p* < 0.03) and increase of HDL-c (*p* < 0.01). No significant changes in the control group.	Intervention BMI: 29.65 ± 0.70 to 29.30 ± 0.72 (*p* = 0.02) Control BMI: 31.86 ± 1.15 to 31.89 ± 1.17	Not significant	Reduction of SBP and DBP (*p* < 0.001) in the intervention group. Control: no significant changes.	There were no adverse effects.	Abedini et al., 2020 [66]
Randomized, triple-blind, placebo-controlled trial with 92 women participants (15–48 years) diagnosed with PCOS.	Iran	Participants received symbiotic pomegranate juice (300 mL/day with symbiotic organisms/8 week) or pomegranate juice (300 mL/day)/8 weeks) or symbiotic beverage (water with symbiotic organisms/8 weeks) or placebo.	SPJ: reduction of LDL-c and increase of HDL-c (*p* < 0.01); PJ: increase of HDL-c (*p* < 0.01); SB: decrease of TC and LDL-c (*p* < 0.01).	Not reported	Not reported	Reduction of SBP and DBP (*p* < 0.001) in the intervention group. Control: no significant changes.	Any adverse effects were encountered.	Esmaeilinezhad et al., 2020 [67]
Randomized crossover trial with 41 hemodialysis participants (47.8 ± 13.3 years, 25♂, 16♀).	Iran	Participants received pomegranate juice (100 mL right after hemodialysis sections 3 x/week) or placebo. This study had a washout design (4-week period), and the study’s intervention took 20 weeks.	Reduction of TG (*p* < 0.001) and increase of HDL-c (*p* < 0.001). No significant changes in the control group.	Not significant	Not reported	Reduction of SBP and DBP (*p* < 0.001) in the intervention group. Control: no significant changes.	Stomach discomfort was reported by 1 dropped-out participant of the pomegranate juice.	Barati Boldaji et al., 2020 [10]
Randomized, single-blind, placebo-controlled clinical trial with 60 participants (40–65 years; 30♀, 30♂) diagnosed with T2DM.	Iran	Subjects were randomly separated into control (*n* = 30) and treatment (*n* = 30, 200 mL of pomegranate juice daily/6 weeks).	Not significant.	Not significant	Not reported	Reduction of SBP and DBP (*p* < 0.001) in the intervention group. Control: no significant changes.	Not reported.	Sohrab et al., 2019 [12]
Randomized study with 23 women (aged 40–60 years) diagnosed with MetS.	Serbia	Participants received 300 mL of polyphenolic-rich pomegranate juice daily/6 weeks) or placebo.	Not significant.	Not significant	Not significant	Not significant.	Not reported.	Kojadinovic et al., 2017 [80]
Randomized, double-blind, placebo-controlled clinical trial with 53 participants (18–65 years, 40♀, 13♂).	United Kingdom	Participants received pomegranate extract capsule (210 mg of punicalagins, 328 mg of polyphenols, and 0.37 mg of anthocyanins)/day/8 weeks) or placebo.	Not reported.	Not significant	Not reported	Reduction of DBP (*p* < 0.05) in the intervention group. Control: no significant changes.	Not reported.	Stockton et al., 2017 [74]
Randomized and controlled clinical trial with 10 healthy participants (5♂ and 5♀).	Greece	Participants received 500 mL of pomegranate juice daily for 14 days or placebo.	Not significant.	Not significant	Not significant	Not significant.	Not reported.	Manthou et al., 2017 [79]
Randomized, double-blind, placebo-controlled trial with 42 (30–60 years) overweight and obese participants.	Iran	Participants received 1000 mg of pomegranate extract containing 40% of ellagic acid daily for 30 days) or placebo.	Intervention: reduction of TC (*p* < 0.003) and LDL-c (*p* < 0.009); increase of HDL-c (*p* < 0.001).	Not significant	Intervention and control: reduction of GLU (*p* < 0.001) and IR (*p* < 0.001)	Not reported.	One individual in the placebo group presented stomach cramps.	Hosseini et al., 2016 [68]
Randomized, double-blind, parallel-group, multicenter, controlled trial with 31 male athletes.	Spain	Participants received placebo or pomegranate juice (200 mL/day) or diluted pomegranate juice (200 mL/day dilute in the same amount of water)/21 day.	PJ and diluted PJ: increase of HDL-c: (*p* < 0.05).	Not significant	Not significant	Not reported.	Not reported.	Fuster-Muñoz et al., 2016 [78]
Prospective, randomized, crossover clinical trial with 24 participants (13♀, 11♂, 61 ± 14 years) diagnosed with ESRN receiving hemodialysis thrice weekly.	USA	Participants received pomegranate juice (100 mL/day/4 weeks) or pomegranate extract (1050 mg/day/4 weeks). After 4 weeks, there was a washout period of 4 weeks and after the intervention was inverted, the groups received the alternative intervention for more 4 weeks.	Not significant.	Not reported	Not reported	Not significant.	No direct study-related adverse effects were reported.	Rivara et al., 2015 [73]
Randomized, placebo-controlled, double-blind with 27 participants that need to pass through hemodialysis.	USA	Participants were randomly assigned into two groups: pomegranate (*n* = 13, 1000 mg capsule of pomegranate’s purified polyphenol extract 7 days/week for 6 months) and placebo (*n* = 14).	Not significant.	Not significant	Not reported	Reduction of SBP and DBP (*p* < 0.05) in the intervention group. Control: no significant changes.	NR here, but the authors suggested no GI adverse effects.	Wu et al., 2015 [70]
Randomized, double-blind, placebo-controlled clinical trial with 101 hemodialysis participants (66.5 ± 11.8 years, 54.5%♂ and 45.5♀).	Israel	Participants received 0.7 mmol of polyphenols in form of 100 cc of pomegranate juice, 3 × weeks/1 year or placebo. The interventions were all made during the first hour of dialysis sections.	Reduction of TG (*p* < 0.05) and increase of HDL-c (*p* < 0.05). No significant changes in the control group.	Not reported	Not reported	Reduction of SBP, DBP, and PP (*p* < 0.05)Control: no significant changes.	NR here, but the authors suggested no GI adverse effects.	Shema-Didi et al., 2014 [81]
Randomized and single-blind clinical trial with 21 participants (30–67 years, 15♀, 6♂) diagnosed with hypertension.	Iran	Participants were randomly separated into intervention group (150 mL daily of pomegranate juice for 2 weeks) and control.	Not significant.	Not reported	Not significant	Reduction of SBP and DBP (*p* < 0.05) in the intervention group. Control: no significant changes.	Not reported.	Asgary et al., 2014 [65]
Randomized, double-blind, placebo-controlled clinical trial with 44 participants (40–60 years, 23♂, 21♀) diagnosed with T2DM.	Iran	Participants were assigned into two groups: intervention (250 mL daily of pomegranate juice containing 1946 mg/L of polyphenols for 12 weeks) and control.	Not significant.	Not significant	Not significant	Not reported.	There were no adverse effects.	Sohrab et al., 2014 [69]
Randomized, double-blind, placebo-controlled clinical trial with 77 overweight women (20–65 years, BMI of 25–35 kg/m^2^).	South Korea	Participants received 200 mL/day of pomegranate vinegar containing 1.5 g of acetic acid added to 700 μg of ellagic acid for 8 weeks) or placebo.	Not significant	Not significant	Not significant	Not reported.	Not reported.	Park et al., 2014 [82]
Randomized, placebo-controlled, crossover clinical trial with 28 participants (16♀, 12♂).	United Kingdom	Participants received 500 mL of pomegranate juice with 1685 mg/L of polyphenols or placebo/4 weeks. After a 1-week washout period, groups were changed to the alternative experiment (placebo group was transformed in intervention and vice-versa)/4 weeks.	Not significant.	Not significant	InterventionHOMA-IR: 2.216 ± 1.43 to 1.825 ± 1.12 (*p* = 0.028) ControlHOMA-IR: 2.245 ± 0.23 to 2.226 ± 0.12 (not significant)	Reduction of SBP and DBP (*p* < 0.031) in the intervention group. Control: no significant changes.	NR here.	Tsang et al., 2012 [76]
Randomized, placebo-controlled, parallel-group, open-label clinical trial with 48 healthy participants (30–50 years, 32♀, ♂16 male).	United Kingdom	The groups were intervention (*n* = 24, 330 mL/day of pomegranate juice for 4 weeks) and control (*n* = 24).	Not reported	Not significant	Not reported	Reduction of SBP and DBP (*p* < 0.001) in the intervention group. Control: no significant changes.	*n* = 2 problems in consumption of the juice.	Lynn et al., 2012 [75]
Randomized, double-blind, placebo-controlled clinical trial with 20 obese participants (25–55 years).	Mexico	The obese patients were randomly assigned into two groups: intervention (120 mL of pomegranate juice daily for 1 month, *n* = 10) and placebo (*n* = 10).	Not significant	InterventionFM (%): 41.3 ± 6.2 to 39.9 ± 6.5 (*p* = 0.010) Control FM (%): 36.3 ± 7.7 to 37.4 ± 7.8	Not significant	Not reported.	There were no adverse effects.	González-Ortiz et al., 2011 [77]
Randomized, double-blind, placebo-controlled study with 45 participants (5♀, 40♂) diagnosed with ischemic coronary heart disease and myocardial ischemia.	USA	Participants were randomly assigned into 2 groups: intervention (240 mL/day of pomegranate juice for 3 months) and control.	Not significant	Not significant	Not significant	Not significant.	Not reported.	Sumner et al., 2005 [71]
Randomized clinical trial with 19 participants (65–75 years, 5♀, 14♀) diagnosed with asymptomatic severe carotid artery stenosis.	USA	Participants received pomegranate juice (50 mL/day of pomegranate with 1.5 mmoles of polyphenols for 1 year) or placebo. After this period, 5 participants of the pomegranate intervention agreed to continue the study for up to 3 years.	Not significant	Not significant	Not significant	Reduction of SBP and DBP (*p* < 0.05) in the intervention group (1 y). Control: no significant changes.	Not reported.	Aviram et al., 2004 [72]

BMI, body mass index (kg/m^2^); cc, cubic centimeters (c^3^); DBP, diastolic blood pressure; ESRN, end-stage renal disease; FM: fat mass; GI, gastrointestinal; GLU, glucose; HDL-c, high-density lipoprotein cholesterol; HOMA, homeostasis model of assessment; IR, insulin resistance; LDL-c, low-density lipoprotein cholesterol; MetS, metabolic syndrome; NR: not reported; PCOS, polycystic ovary syndrome; PJ, pomegranate juice; PP, pulse pressure; SB, symbiotic beverage; SBP, systolic blood pressure; SPJ, symbiotic pomegranate juice; TC, total cholesterol; TG; triglycerides; T2DM, type 2 diabetes mellitus.

**Table 3 nutrients-14-01665-t003:** Possible biases of the included randomized clinical trials (RCTs).

Study	Question Focus	Appropriate Randomization	Allocation Blinding	Double-Blind	Losses (<20%)	Prognostic and Demographic Characteristics	Outcomes	Intention to Treat Analysis	Sample Calculation	Adequate Follow-Up
Abedini et al., 2020 [66]	Yes	No	No	No	Yes	Yes	Yes	No	Yes	Yes
Esmaeilinezhad et al., 2020 [67]	Yes	Yes	Yes	Yes	Yes	Yes	Yes	No	Yes	Yes
Barati Boldaji et al., 2020 [10]	Yes	Nr	No	No	Yes	Yes	Yes	Yes	Yes	Yes
Sohrab et al., 2019 [12]	Yes	Yes	No	No	Yes	Yes	Yes	No	Nr	Yes
Kojadinovic et al., 2017 [80]	Yes	Nr	No	No	Yes	Yes	Yes	Yes	Nr	Yes
Stockton et al., 2017 [74]	Yes	Yes	Yes	Yes	Yes	Yes	Yes	No	Yes	Yes
Manthou et al., 2017 [79]	Yes	Nr	No	No	Yes	Yes	Yes	Yes	Yes	No
Hosseini et al., 2016 [68]	Yes	No	Yes	Yes	Yes	Yes	Yes	No	Yes	Yes
Fuster-Muñoz et al., 2016 [78]	Yes	Nr	Yes	Yes	No	Yes	Yes	No	No	No
Rivara et al., 2015 [73]	Yes	Nr	No	No	Yes	Yes	Yes	Yes	Nr	Yes
Wu et al., 2015 [70]	Yes	Nr	Yes	No	Yes	Yes	Yes	No	Nr	Yes
Shema-Didi et al., 2014 [81]	Yes	Nr	Yes	Yes	No	Yes	Yes	No	Nr	Yes
Asgary et al., 2014 [65]	Yes	Nr	No	No	Yes	Yes	Yes	Yes	Nr	No
Sohrab et al., 2014 [69]	Yes	No	Yes	Yes	Yes	Yes	Yes	No	Yes	Yes
Park et al., 2014 [82]	Yes	Yes	Yes	Yes	Yes	Yes	Yes	No	Yes	Yes
Tsang et al., 2012 [76]	Yes	Nr	No	No	Yes	Yes	Yes	Yes	Nr	Yes
Lynn et al., 2012 [75]	Yes	No	No	No	Yes	Yes	Yes	No	Yes	Yes
González-Ortiz et al., 2011 [77]	Yes	Nr	Yes	Yes	Yes	No	Yes	Yes	Yes	Yes
Sumner et al., 2005 [71]	Yes	No	Yes	Yes	Yes	Yes	Yes	No	Nr	Yes
Aviram et al., 2004 [72]	Yes	Nr	No	No	Yes *	No	Yes	Yes *	Nr	Yes

Nr, not reported. * This study had initially 19 participants. After the first year of treatment, only 5 participants continued the study intervention for three years. The ‘’yes’’ is for the 1-year original intervention.

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
