# Peer review of "Pomegranate (Punica granatum L.) and Metabolic Syndrome Risk Factors and Outcomes: A Systematic Review of Clinical Studies"

_nutrients, 2022, doi:10.3390/nu14081665_

Round 1
Reviewer 1 Report
This is an appropriately conducted SLR, that followed and reported as per PRISMA. I commend authors for such a detail and comprehensive review. I have provided a few minor suggestions below:
- Plesae consider using MetS as the short form for metabolic syndrome since MS can be confusing with Multiple Sclerosis.
- In Table 2 column Reference please consider adding the Author last name and year, eg. Abedini et al 2020
- Similar as above can be done for Table 2
Author Response
The authors of this manuscript express their sincere thanks to the Section Managing Editor and the reviewer for the critical assessment of this work. The authors have acted upon the recommendations of the Section Managing Editor and the reviewer which have resulted in a significant enhancement in the quality of this manuscript. All modifications incorporated in the manuscript are highlighted in red color font. A “point-by-point” response to each and every comment is outlined below.
General comments:
This is an appropriately conducted SLR, that followed and reported as per PRISMA. I commend authors for such a detail and comprehensive review. I have provided a few minor suggestions below:
Response:
We would like to extend our gratitude to the reviewers for providing their valid comments. Our point-by-point response for each comment is provided below.
Specific comments
Comment 1:
Please consider using MetS as the short form for metabolic syndrome since MS can be confusing with Multiple Sclerosis.
Response:
This is an excellent suggestion. We have replaced “MS” with “MetS” in the entire manuscript.
Comment 2:
In Table 2 column Reference please consider adding the Author last name and year, eg. Abedini et al 2020.
Response:
Thank you very much for your suggestion. We included the names of the authors for all references in Table 2 (pages 11-16).
Comment 3:
Similar as above can be done for Table 2
Response:
We assume the reviewer meant Table 3. We included the names of the authors for all references in Table 3 (pages 15-17).
Additionally,
The entire manuscript has been thoroughly checked and edited to minimize typographical errors as well as to ensure uniform style, organization, and quality.
Finally,
On behalf of my co-authors, I once again express my sincere thanks to the erudite Associate Editor and reviewers for the valuable suggestions and constructive input to improve the quality of our manuscript.
Reviewer 2 Report
Thank you for giving me the opportunity to review this piece of work. In this work, Laurindo et al. carried out a systematic review of literature on pomegranate intake and its effect on metabolic syndrome and its outcomes.
Kindly, find below my comments for your response.
Abstract
Line 22: The authors must state the “date” the search was carried out. The authors should indicate how many articles were retrieved after the search and state how many were selected after the removal of duplicates and screening for eligibility.
Line 19: The authors must be very clear what their outcome of interest is. In some instances, they state that, its “risk factors of metabolic syndrome” as indicated for instance at Line 19 whereas in some cases they indicate “metabolic syndrome parameters” for example at Line 21. This is important because Obesity for instance is a potential risk factor for metabolic syndrome development whereas dyslipidaemia, impaired glucose metabolism are outcomes of metabolic syndrome.
Introduction:
The authors consistently cite several references together for a bulky paragraph. That is not ideal. The authors must place each reference by the each supporting statement.
Line 45: replace “most” with “mostly”
Line 48: the authors should revise to “…..P. granatum consumption…….”
- Literature search strategy and study selection
The authors must state the date the article extraction was carried out and by how many of the authors stated on the manuscript. The authors must thoroughly explain the entire article search process that yielded the finally selected articles. The authors should state which databases were used, how many duplicate papers were removed, how many articles were obtained following the screening of title and abstract, assessment of eligibility and following the application of the “inclusion” and “exclusion” criteria. That will be the explanation of the PRISMA guideline.
Line 113-115: That statement there is not relevant. The authors must go straight into the Method employed for the article retrieval.
Line 116: The databases listed there are not all captured in those stated for the “Abstract”. The authors must include all the databases stated here in the “Abstract”.
Line 117-120: For the descriptors indicated, I can see that the authors outcome of interest were both “risk factors of metabolic syndrome” and “metabolic disease outcomes” as well.
Line 121: The authors state that there was a use of “filter”. However, they fail to state where the filtering was done. Did they filter the search outcome based on “years of publication” for example? Or they filtered based on the type of articles retrieved?
Line 123-124: The authors used the older version of the PRISMA guideline by Liberati et al. (2009). However, there is a newly updated PRISMA guideline by (Page et al. 2021) which is what the authors must use.
Line 125 -127: The inclusion criteria stated there is not exhaustive. In Line 15, the authors selection of articles published only in English is also an inclusion criteria and must be added as such.
Line 125: RCT is the abbreviation for “Randomised Controlled Trials and not “Randomised Clinical Trials”
Line 128: revise “comprised in” to “comprised of”
Line 130: revise “risk bias” to “risk of bias”
- Overview of selected clinical studies
I think this section should be changed from “Overview of selected clinical studies” to “Result findings”.
Line 135-139: The authors have indicated the countries those studies were conducted from. This can rather be under the “Discussion” section.
Line 139-142: The authors must indicate the “references” for each of the food formulations indicated there.
In the PRISMA guideline presented above, the authors must indicate the number of articles obtained from each of the databases “Medline, Google Scholar, Lilacs, PubMed, Embase, Cochrane, and 116 Clinical Trials.gov” listed in the preceding section above.
Also, still on the PRISMA guideline presented, I don’t get why the authors have indicated “Articles included in the quantitative synthesis”. Articles included in quantitative synthesis are for “Meta-analysis”. Yet, in this review, the authors did not conduct a “Meta-analysis”.
Line 155-158: This is a “Discussion”.
Table 2: The authors must revise the Table to include the “Age” of participants, their “BMI or weight”. The section heading “Patient” is not right. The authors must split that into “Country of study” and “Type of study”. The authors must state the objectives of each study.
Table 3. must be revised to capture these columns “random sequence generation, allocation concealment, blinding of participants or study personnel, blinding of outcome assessment, incomplete outcome data, selective reporting, and other bias”.
- Clinical trials showing the effects of P. granatum on MS
This section title must be changed to “Discussion”.
These section titles “5. Bioavailability of P. granatum” and “6. Safety of P. granatum” must all be removed and the discussions under them added to the preceding discussion.
Under the discussion on the “Safety of P. granatum”, the authors discuss the toxicological study using animal models. Yet, in the inclusion criteria used for the article selection, there was no mention of “Toxicological studies” in animal models. The authors adding this discussion to the narrative makes the review a blend of systematic review and narrative review as well. For the sake of the overarching goal of the review focused on a systematic review, the authors must stick to that and only discuss papers that were selected for the systematic review.
Conclusion
Because the authors in their string of search terms used terms related to “risk factors” as well as the “metabolic syndrome outcomes”. Therefore, it will be a great idea to tweak the title to capture the “risk factors” and conclude on that as well.
Author Response
The authors of this manuscript express their sincere thanks to the Section Managing Editor and the reviewer for the critical assessment of this work. The authors have acted upon the recommendations of the Section Managing Editor and the reviewer which have resulted in a significant enhancement in the quality of this manuscript. All modifications incorporated in the manuscript are highlighted in red color font. A “point-by-point” response to each and every comment is outlined below.
General comments:
Thank you for giving me the opportunity to review this piece of work. In this work, Laurindo et al. carried out a systematic review of literature on pomegranate intake and its effect on metabolic syndrome and its outcomes.
Response:
We would like to extend our gratitude to the reviewers for providing their valid comments. Our point-by-point response for each comment is provided below.
Specific comments
Abstract
Comment:
Line 22: The authors must state the “date” the search was carried out. The authors should indicate how many articles were retrieved after the search and state how many were selected after the removal of duplicates and screening for eligibility.
Response:
Thank you very much for your comment. We included a date for the literature search and explain details on the literature search and the selection of appropriate studies (page 1, lines 25-27). We also modified the PRISMA flowchart (Figure 2) according to updated PRISMA guidelines published by Page et al., 2021.
Comment:
Line 19: The authors must be very clear what their outcome of interest is. In some instances, they state that, its “risk factors of metabolic syndrome” as indicated for instance at Line 19 whereas in some cases they indicate “metabolic syndrome parameters” for example at Line 21. This is important because Obesity for instance is a potential risk factor for metabolic syndrome development whereas dyslipidaemia, impaired glucose metabolism are outcomes of metabolic syndrome.
Response:
This is a very nice suggestion. We changed “metabolic syndrome parameters” to “metabolic syndrome risk factors” since it is more appropriate (page 1, line 21; page 2, line 75; page 8, line 114; and page 9, line 171).
Introduction
Comment:
The authors consistently cite several references together for a bulky paragraph. That is not ideal. The authors must place each reference by the each supporting statement.
Response:
Thank you very much for your comment. We have cited appropriate references according to each statement (page 1, lines 39, 40 and 43; and page 2, lines 46, 51, 53, 58, 62, 66-67, 71, 74, 78 and 81-85).
Comment:
Line 45: replace “most” with “mostly”
Response:
The word “most” has been replaced with “mostly” (page 2, line 48).
Comment:
Line 48: the authors should revise to “…..P. granatum consumption…….”
Response:
Thank you very much for your comment. We modified the sentence according to your suggestion (page 2, line 51).
Literature search strategy and study selection
Comment:
The authors must state the date the article extraction was carried out and by how many of the authors stated on the manuscript. The authors must thoroughly explain the entire article search process that yielded the finally selected articles. The authors should state which databases were used, how many duplicate papers were removed, how many articles were obtained following the screening of title and abstract, assessment of eligibility and following the application of the “inclusion” and “exclusion” criteria. That will be the explanation of the PRISMA guideline.
Response:
These are very nice suggestions. We have included the date when the extraction was performed and how many authors did the search (page 8, lines 116-117).
Moreover, we also included the databases that were used (page 8, lines 119-120).
Additionally, we have included how many studies were identified in the databases, how many were removed after duplications, and how many remained after using the inclusion and exclusion criteria (page 8, lines 137-141).
Comment:
Line 113-115: That statement there is not relevant. The authors must go straight into the Method employed for the article retrieval.
Response:
We are in absolute agreement with the reviewer. We have removed the sentence “Literature search is important to identify the appropriate design of the study, methodology, population studied, and appropriate methods. It also can help identify variables that could affect the outcomes.”
Comment:
Line 116: The databases listed there are not all captured in those stated for the “Abstract”. The authors must include all the databases stated here in the “Abstract”.
Response:
We agree with the reviewer’s comment. Accordingly, we included appropriate information both in the abstract (page 1, lines 19-20) and in the text (page 8, lines 119-120).
Comment:
Line 117-120: For the descriptors indicated, I can see that the authors outcome of interest were both “risk factors of metabolic syndrome” and “metabolic disease outcomes” as well.
Response:
We appreciate this comment. We modified it for “components of MetS” (page 8, line 126).
Comment:
Line 121: The authors state that there was a use of “filter”. However, they fail to state where the filtering was done. Did they filter the search outcome based on “years of publication” for example? Or they filtered based on the type of articles retrieved?
Response:
We removed the word “filter” and reformulated the sentence: “All these descriptors helped the authors to identify published clinical trials that associated the consumption of any part or extract of P. granatum or…” (page 8, lines 123-124).
Comment:
Line 123-124: The authors used the older version of the PRISMA guideline by Liberati et al. (2009). However, there is a newly updated PRISMA guideline by (Page et al. 2021) which is what the authors must use.
Response:
We are grateful to the reviewer for this comment. We have indicated the use of newly updated PRISMA guidelines in the text (page 8, lines 127-128). We have also revised the PRISMA flowchart accordingly (Figure 2, page 9).
Comment:
Line 125 -127: The inclusion criteria stated there is not exhaustive. In Line 125, the authors selection of articles published only in English is also an inclusion criteria and must be added as such.
Response:
Thank you very much for your comment. We added the articles published in English as an inclusion criteria (page 8, line 131).
Comment:
Line 125: RCT is the abbreviation for “Randomised Controlled Trials and not “Randomised Clinical Trials”.
Response:
We corrected the mistake (page 8, line 128).
Comment:
Line 128: revise “comprised in” to “comprised of”
Response:
We have rectified the error (page 8, line 131).
Comment:
Line 130: revise “risk bias” to “risk of bias”
Response:
We have changed “risk bias” to “risk of bias” (page 8, line 133).
Overview of selected clinical studies
Comment:
I think this section should be changed from “Overview of selected clinical studies” to “Result findings”.
Response:
We modified the section title according to your suggestion (page 8, line 136).
Comment:
Line 135-139: The authors have indicated the countries those studies were conducted from. This can rather be under the “Discussion” section.
Response:
While we appreciate this comment, we think if we include the countries in the “Discussion section”, there would be a repetition of the results.
Comment:
Line 139-142: The authors must indicate the “references” for each of the food formulations indicated there.
Response:
We have included the references as suggested (page 8, lines 146-149).
Comment:
In the PRISMA guideline presented above, the authors must indicate the number of articles obtained from each of the databases “Medline, Google Scholar, PubMed, Embase, Cochrane, and 116 Clinical Trials.gov” listed in the preceding section above.
Response:
We have revised the PRISMA flowchart according to your previous suggestion (page 9, Figure 2).
Comment:
Also, still on the PRISMA guideline presented, I don’t get why the authors have indicated “Articles included in the quantitative synthesis”. Articles included in quantitative synthesis are for “Meta-analysis”. Yet, in this review, the authors did not conduct a “Meta-analysis”.
Response:
We admire the reviewer for this critical comment. We have modified the PRISMA flowchart (page 9, Figure 2).
Comment:
Line 155-158: This is a “Discussion”.
Response:
We modified the sentence as “The included studies have shown that the treatment with pomegranate, pomegranate extracts, or pomegranate-derived substance was able to improve cardiometabolic risk factors, such as fasting blood glucose and glycemic levels, systolic and diastolic blood pressures, lipid profiles, and body weight, which are the main diagnostic aspects of MetS (Table 2).” (page 9, lines 163-168).
Comment:
Table 2: The authors must revise the Table to include the “Age” of participants, their “BMI or weight”. The section heading “Patient” is not right. The authors must split that into “Country of study” and “Type of study”. The authors must state the objectives of each study.
Response:
This is an excellent suggestion. Accordingly, we have revised the table (page 11) as follows:
a) We modified the heading of the first column of the table as “Type of the study”;
b) We have also modified the heading of the second column as “Country of the study”;
c) If available, the age of the patients of the included studies was mentioned in the first column of the table and the body weight was included in the fifth column.
Comment:
Table 3. must be revised to capture these columns “random sequence generation, allocation concealment, blinding of participants or study personnel, blinding of outcome assessment, incomplete outcome data, selective reporting, and other bias”.
Response:
Thank you very much for your comment. We followed the model of the Cochrane Handbook for Systematic Reviews of Interventions (Higgins, J.P.; Thomas, J.; Chandler, J.; Cumpston, M.; Li, T.; Page, M.J.; Welch, V.A. Cochrane handbook for systematic reviews of interventions; John Wiley & Sons: 2019), for this reason, we do not think it is necessary to include other biases.
Clinical trials showing the effects of P. granatum on MS
Comment:
This section title must be changed to “Discussion”.
Response:
We have changed this section title according to your suggestion (page 19, line 199).
Comment:
These section titles “5. Bioavailability of P. granatum” and “6. Safety of P. granatum” must all be removed and the discussions under them added to the preceding discussion.
Response:
This is an excellent suggestion. We have removed this section and included the text in the Discussion section as appropriate (page 19, lines 201-217).
Comment:
Under the discussion on the “Safety of P. granatum”, the authors discuss the toxicological study using animal models. Yet, in the inclusion criteria used for the article selection, there was no mention of “Toxicological studies” in animal models. The authors adding this discussion to the narrative makes the review a blend of systematic review and narrative review as well. For the sake of the overarching goal of the review focused on a systematic review, the authors must stick to that and only discuss papers that were selected for the systematic review.
Response:
We are in absolute agreement with the reviewer, and we removed this section from the paper and included it in the Discussion section (page 19, lines 201-217).
Conclusion
Comment:
Because the authors in their string of search terms used terms related to “risk factors” as well as the “metabolic syndrome outcomes”. Therefore, it will be a great idea to tweak the title to capture the “risk factors” and conclude on that as well.
Response:
We applaud the reviewer for this thought-provoking comment. We have modified the title to “Pomegranate (Punica granatum L.) and Metabolic Syndrome Risk Factors: A Systematic Review of Clinical Studies” (page 1, lines 1-2). We have also revised the conclusion section as well (page 23, line 391).
Additionally,
The entire manuscript has been thoroughly checked and edited to minimize typographical errors as well as to ensure uniform style, organization, and quality.
Finally,
On behalf of my co-authors, I once again express my sincere thanks to the erudite Associate Editor and reviewers for the valuable suggestions and constructive input to improve the quality of our manuscript.
Round 2
Reviewer 2 Report
Thanks for undertaking the revision. The manuscript quality has improved. However, I have still got some comments. Kindly find below some minor comments for your action.
Title:
Revise the title to “Pomegranate (Punica granatum L.) and Metabolic Syndrome Risk Factors and Outcomes: A Systematic Review of Clinical Studies”. This is because from the review, there are metabolic syndrome “outcomes” as well as the “risk factors”.
Abstract:
Line 23: The authors should state the date (Day, month and year) the search was conducted.
In the new PRISMA guideline, records identified from Registers should be reported. The authors have indicated that some articles were obtained from “Clinical Trials.gov”. That appears to me as a “Register”.
Also, just out of curiosity, did the authors search “EMBASE, MEDLINE and Clinical Trials.gov” separately or they obtained the results from “Cochrane Library?”
- Literature search strategy and study selection
The authors should again state the date (Day, month and year) the search was conducted. Also, the name initials of the two authors who undertook the search for the literature should be indicated.
- Result findings
In the PRISMA guideline, the authors should indicate the total number of articles obtained from each search on each of the databases that was searched. For example Google scholar= 2xxxx, EMBASE= 31xxxxxx….
Discussion
The first paragraph of the discussion is not right. The authors should discuss the results or findings of the studies selected for the qualitative meta-analysis. Those in vitro studies the authors are discussing are not part of the inclusion criteria used for the selection of the articles.

Author Response
The authors of this manuscript express their sincere thanks to the reviewer for the critical assessment of this work. The authors have acted upon the recommendations of the Section Managing Editor and the reviewers which have resulted in a significant enhancement in the quality of this manuscript. All modifications incorporated in the manuscript are highlighted in red color font. A “point-by-point” response to each and every comment is outlined below.
General comments:
Thanks for undertaking the revision. The manuscript quality has improved. However, I have still got some comments. Kindly find below some minor comments for your action.
Response:
Thank you very much for your time in reviewing our revised manuscript. We appreciate the minor comments that we carefully considered while revising our manuscript.
Specific comments:
Title:
Comment:
Revise the title to “Pomegranate (Punica granatum L.) and Metabolic Syndrome Risk Factors and Outcomes: A Systematic Review of Clinical Studies”. This is because from the review, there are metabolic syndrome “outcomes” as well as the “risk factors”.
Response:
We are in absolute agreement with this excellent suggestion and accordingly modified the title.
Abstract:
Comment:
Line 23: The authors should state the date (Day, month and year) the search was conducted.
Response:
Thank you very much for your comment. We included the day, month, and year of the search (page 1, line 24).
Comment:
In the new PRISMA guideline, records identified from Registers should be reported. The authors have indicated that some articles were obtained from “Clinical Trials.gov”. That appears to me as a “Register”.
Response:
The reviewer is absolutely correct. We indicated the number of articles obtained from “Clinical Trials.gov” in the revised PRISMA flowchart (Figure 2, page 9).
Comment:
Also, just out of curiosity, did the authors search “EMBASE, MEDLINE and Clinical Trials.gov” separately or they obtained the results from “Cochrane Library?”
Response:
We performed the search separately.
Literature search strategy and study selection
Comment:
The authors should again state the date (Day, month and year) the search was conducted. Also, the name initials of the two authors who undertook the search for the literature should be indicated.
Response:
We are grateful for the suggestion. We included the correct date and the initials of the authors who performed the search (page 8, lines 119 and 120)
Result findings
Comment:
In the PRISMA guideline, the authors should indicate the total number of articles obtained from each search on each of the databases that was searched. For example Google scholar= 2xxxx, EMBASE= 31xxxxxx….
Response:
Thank you for the excellent suggestion. We have revised the PRISMA figure (Figure 2, page 9). and included the number of the articles that were obtained in each database
Discussion
Comment:
The first paragraph of the discussion is not right. The authors should discuss the results or findings of the studies selected for the qualitative meta-analysis. Those in vitro studies the authors are discussing are not part of the inclusion criteria used for the selection of the articles.
Response:
We agree with this suggestion. We removed the first paragraph from the “Discussion” section (page 17).
On behalf of my co-authors, I once again express my sincere thanks to the erudite Associate Editor and reviewers for the valuable suggestions and constructive input to improve the quality of our manuscript.